# It's about Time: Film, Video Games, and the Advancement of an Artform

**Steven Gimbel [1,]*** **and Joseph Roman [2]**

[1]    Department of Philosophy; Gettysburg College, Gettysburg, PA 17325, USA
[2]    Department of Philosophy, University of Illinois, Champaign-Urbana, Champaign, IL 61820, USA; jaroman3@illinois.edu
*    Correspondence: sgimbel@gettysburg.edu

**Abstract:** Jon Robson and Aaron Meskin have argued that the insights obtained through the philosophical analysis of video games is not specific to video games, but to a larger class of artistic creations they term Self-Involving Interactive Fictions, or SIIFs. But there is at least one aspect of SIIF video games that is philosophically interesting and does not apply to the class of SIIFs as a whole, the ability to represent non-classical time. If SIIF video games are considered to be an extension of the art form of graphic narrative story-telling, the art form dominated by film, then the ability to represent time in in this fashion represents a revolution akin to that of vanishing point perspective in painting. This makes SIIF video games philosophically interesting for both philosophers of film and philosophers of video games.

**Keywords:** time; art; video games; relativity

---

Johan Huizinga [1], in his seminal book *Homo Ludens*, contends that while play predates humanity, the way we play for fun is what makes us the beings we are. The sorts of games we develop and engage in for leisure tell us about the essence of our humanity. Following Martin Heidegger, 20th century French phenomenologists like Paul Ricoeur, Giles Deleuze, and Alain Badiou countered Huizinga, contending that it is narrative that is the essential component of our humanity. We are shattered beings, broken into uncountably many individuated bits of memory, and it is only though the use of narrative to impose order and meaning upon them that we become coherent selves. As such, literature and film became philosophical artifacts in need of unpacking. Kendall Walton [2], in *Mimesis as Make-Believe*, brings these two traditions together, contending that we must understand narrative as a form of play. A certain category of video game is at its core an integrally interwoven combination of play and narrative. So, it may seem, and we want to argue that it is the case, that these narrative-based video games should in themselves be a deeply philosophically interesting phenomenon whose intrinsic properties need to be studied to understand how they disclose our humanity.

But this view is opposed by contemporary writers like Florian Cova and Amanda Garcia, and Jon Robson and Aaron Meskin. Cova and Garcia see an important difference between play and narrative. Play, Huizinga notes, is not only free; it is, he contends, freedom itself. To play is to rid yourself of the constraints of society, indeed, even the constraints of reality. A child can make believe that she is flying, that she is four people, that she can violate the principle of the excluded middle while going faster than the speed of light and remaining constant in mass, a mass that obeys a gravitational law which is subject to an inverse cube relation and which possesses field values that change instantaneously through all of space. But, Cova and Garcia point out, this freedom is not observed in narrative. The canonical approach to narrative has a beginning, a middle, and—most importantly for them—a single end. The overwhelming majority of examples of narrative have a single ending. If narrative is essential to humanness, then this is an important fact that speaks against Walton's understanding of narrative as play.

It is true, they point out, that there are examples of multiply ending or even open-ended narratives, but these are exceptional to the point that they reinforce the rule. There are plays that include audience members as characters and therefore whose ending will be unscripted. There are DVDs that include alternative endings for films. There are choose-your-own-adventure books where there is a branching architecture built into the work. And, they point out, there are narrative-based video games. These, especially the video games, show that the linearity and deterministic elements of canonical narrative are not constraints of technology [3] (p. 113). Narratives with multiple endings could become the norm, but they have not. The canonical approach remains the near universal standard. This, they argue, should tell us something deep about our psychology.

Robson and Meskin object that there is a class of video games that are indicative of something important and do warrant serious philosophical consideration. But it is not because these video games are themselves interesting qua video games. Rather, it is because they are the most straightforward examples of a larger class of narratives, Self-Involving Interactive Fictions, or SIIFs.

Not all video games are SIIFs. *Tetris*, for example, is not [4] (p. 166). Nor is it true that all SIIFs are video games. Again, choose-your-own adventure books or tabletop role playing games (RPGs) like *Dungeons and Dragons* are also examples of SIIFs. It is the notion of the SIIF, not the video game that may be of philosophical import.

> "while it is unlikely that video games are what Dominic Lopes has called an 'appreciative art kind', there is at least some case to be made that SIIFs are. Lopes characterizes an appreciative art kind as a group of art works in which we 'normally appreciate a work in the kind by comparison with arbitrarily any other works in that kind'. And this is not true of video games. Many video games—for example, computerized games of chess—do not seem to qualify as artworks at all. Further, even restricting our attention to those games which are more plausibly categorized as artworks, some comparisons still seem problematic. It is implausible, for example, that non-fiction video games such as *Tetris* are normally compared in the appropriate ways to *Bioshock* or *Resident Evil*. By contrast, comparisons between tabletop RPGs and video games are already commonplace". [4] (p. 174)

The category of video games is likely an umbrella term that should be considered a reed that can support little philosophical weight, but the notion of an SIIF, on the other hand, may be considerably more important.

> "we think that a number of the most significant contributions made in recent years to the literature on the philosophy of video games are best considered not as claims about video games but, rather, as claims concerning SIIFs". [2] (p. 174)

While it may be true that SIIFs writ large are a philosophically interesting category and that video games, in such discourse, may be used as simple examples of the broader class of SIIFs, we contend that there is something philosophically and aesthetically important about video game SIIFs qua video games, something that is not true of the broader class of SIIFs. Narrative video games of this sort potentially represent a revolutionary advancement in graphical narrative art.

Artforms are constrained by the nature of their media. There may be, however, technical advances that allow artists to transcend these constraints. Painting, for example, is constrained to two dimensions, but the development of vanishing point perspective revolutionized the artform allowing for three-dimensional representation. For this reason, the development of vanishing point perspective may be considered to be the most important technical development in the history of painting. We contend that a similar revolution is now occurring in the art of animated projected storytelling. Where vanishing point perspective was a development in the artists' ability to represent space, this development the video game SIIF radically alters the artists' ability to represent the dimension of time.

The artform most commonly associated with the artistic manipulation of time is film. A painting, a photograph, or a statue captures what physicists call an "event", that is, a place at a time. But moving

pictures allow for the representation of a place, or places, across time. The nature of this time in film, however, is constrained to be classically Newtonian.

Albert Einstein led a revolution in physics with his theories of relativity that forced scientists to reconceptualize time. This Einsteinian relativistic time, because of the structure of the medium, is inaccessible to the medium of film. Film-makers, while they may seek to present relativistic time in the content of their narratives, are stuck artistically in classical Newtonian time because of the nature of their medium. It is, however, possible to create graphical narrative artworks that are temporally relativistic not only in content, but also in form; but to do so, it requires a different medium of graphical narrative art, it requires the SIIF video game. These games are different from films and different from other SIIFs like choose-your-own adventure books and tabletop RPGs in being capable of becoming artworks with intrinsically relativistic time elements.

As vanishing point perspective completely revolutionized painting, allowing it to transcend the representational limits of a physical dimension constrained by its medium, the move from film to SIIF video game has similarly revolutionized the artform of animated projected storytelling. For this reason, contra Robson and Meskin, there is something intrinsically philosophically important about video game SIIFs.

## 1. Types of Time

To make this case, we must be careful to clearly set out what is meant by time. Indeed, there are four distinctions that need to be kept distinct: (1) film time vs. plot time, (2) physical time vs. phenomenological time, (3) Newtonian classical time vs. Einsteinian relativistic time, and (4) McTaggart's A-time vs. B-time.

### 1.1. Plot Time vs. Film Time

There are two quite different times present in film. There is the time of the film and the time of the story being told in the film. Plot time is the represented time within the content of the film. If the film is narrative, then there are events that are ordered in time. That ordering is plot time. This is to be contrasted with film time which is the time elapsed in the presentation of the images that constitute the film. If a film begins with a flashforward, then what is seen at, say, second 30 of film time may be two years later in plot time.

Altering plot time is a standard narrative strategy. In a 90-min running film time, the arc of the plot could cover 500 years or thirty seconds. In a story told non-linearly, the advance of the film time might jump around in plot time. The central narrative device of *Pulp Fiction* and *Slumdog Millionaire* is to disclose the interlocking causally-related events out of causal order, allowing you to know what happened before and what happened after, but the mystery being how those events connect. Plot time may be explained to obey laws different from film time, perhaps it goes backward for some or all characters or perhaps it stops for some, but not all characters as in *The Curious Case of Benjamin Button*. Perhaps the plot time has a circular, instead of linear structure. The classic horror film *Dead of Night* ends with a scene that is identical to the film's opening scene indicating that the universe of the film is a version of Nietzsche's eternal recurrence. It is the filmmaker's creative prerogative to determine how to present the passing of time, the way characters experience time, and indeed the nature of the time of the film's universe. Film-makers are god-like in having the ability to not only program the events of their fictional universe, but also the laws that those events obey. This is plot time.

But our interest in time is not plot time, rather it is with the seemingly more mundane film time. The film is the ordering of images to be shown over a period of time. You can order those images in time to tell a story with all sorts of complex non-linear plot times, but the ordering of the images itself is the film time. The running time of the film is a measure of the duration of the film time. Film time is the medium of the film-maker. Film-making is the art of changing images over time. It is this time beneath the images that we term "film time" and is the crucial one for our argument.

## 1.2. Physical Time vs. Phenomenological Time

A second distinction made explicitly in the writings of physicist/mathematician/philosopher Henri Poincaré is between physical and phenomenological time. Phenomenological time is the internal experience of time passing.

> "So long as we do not go outside the domain of consciousness, the notion of time is relatively clear. Not only do we distinguish without difficulty present sensation from the remembrance of past sensations or the anticipations of future sensations, but we know perfectly well what we mean when we say that, of two conscious phenomena which we remember, one was anterior to the other; or that, of two foreseen conscious phenomena, one will be anterior to the other.
>
> When we say that two conscious facts are simultaneous, we mean that they profoundly interpenetrate, so that analysis cannot separate them without mutilating them.
>
> The order in which we arrange conscious phenomena does not admit of any arbitrariness. It is imposed upon us and of it we can change nothing". [5] (p. 26)

Phenomenological time is the internal time of our experience, but physical time is the time of the processes in the universe independent of the experiences of any consciousness.

> "[I]nto this form we wish to put not only the phenomena of our own consciousness, but those of which other conscisounesses are the theater. But more, we wish to put there physical facts, these I know not what with which we people space and which no consciousness sees directly. This is necessary because without it science could not exist". [5] (pp. 26–27)

We all know the phenomenon of the divergence of phenomenological and physical times. Time flies when you are having fun. Physical time passes at the same rate, but the rate of experienced time changes with the level of enjoyment of the activity that fills that time. A boring class and an exciting film may both take ninety minutes, but the length of those identical physical durations will be experienced very differently. Our interest here is in physical time of the film.

## 1.3. Newtonian Classical Time vs. Einsteinian Relativistic Time

Our understanding of physical time comes from physics. As our physical theories change, our concepts change accordingly. As such, with the Einsteinian revolution in physics, came a radical change in how we understand time.

The classical or Newtonian conception of time derives from the theory of mechanics set out by Isaac Newton in his masterwork *The Mathematical Principles of Natural Philosophy* [6]. At the beginning of this book, Newton starts by defining his basic concepts. At the end of these definitions, he includes an addendum, a Scholium, in which he informs the reader that there are four concepts he does not define because "they are well-known to all". These four notions that need no definition are space, place, motion, and most importantly for us, time.

Each of these notions have two senses, one that is merely "relative, apparent, and common", and another that is "absolute, true, and mathematical". In terms of the real meaning of physics, it is clearly the latter that is of importance. We are not interested in the relative measure of time, Newton contends, but rather in the absolute, true time itself.

Of this, Newton writes,

> "Absolute, true, and mathematical time, of itself and from its own nature, flows equably without relation to anything external, and by another name is called duration: relative, apparent, and common time is some sensible and external (whether accurate or unequable) measure of the duration by means of motion, which is commonly used instead of true time; such as an hour, a day, a month, a year". [6] (p. 6)

Watches and clocks, by means of motion, tell us the relative time. But philosophically, to truly understand the nature of reality itself, we must move beyond the motion of the sun in the sky, the motion of a hand across a watch face, the numbers on the front of the microwave oven, and get to the underlying time itself.

The best way to explain Newton's notion of absolute time is through a metaphor and the best model to use for this just happens to be film. Picture a film. There is a linear string of frames printed on a strip of celluloid. Each individual frame constitutes the space of the universe. For Newton, space, like time, is absolute. This means that there is a fact of the world about the location of objects. For any object in a frame, we can measure the distance of it across and up from the bottom left-hand corner of the frame. Space and place are fixed and absolute.

These frames are printed in an order. That order is the time and it is fixed and absolute. The absolute nature of time manifests itself in three ways. First, there is absolute simultaneity. It is a fact of the world whether two events are simultaneous, that is, whether or not they appear in the same frame. Secondly, there is absolute time order, that is, there are fixed and objective facts of the world whether one event is earlier than (appears in an earlier frame) or later than (appears in a later frame) another event. Third, there is absolute duration, that is, we can count the frames between two events to give an absolute and objective measure of how long between events. Newton's time flows in one direction at a constant rate that creates objective facts of the world about whether two things happened at the same time, earlier or later, and how long it took. This is a rigorous working out of what is the commonsense understanding of time.

It was undermined by Albert Einstein's theory of special relativity, first published in his article "On the Electrodynamics of Moving Bodies" [7] in 1905. The theory of Newton, which nearly perfectly accounted for the motion of planets around the sun and the falling of apples to the Earth, had been showed to be irreconcilable with James Clerk Maxwell's theory that accounted for electricity, magnetism, and light. Since the complete understanding of the universe would have to be coherent, we could not accept both of these theories. While Einstein's contemporaries worked hard to change Maxwell's work to make it consistent with Newton, Einstein took the opposite approach and changed Newton's theory to make it consistent with Maxwell. The idea of replacing Newton was unthinkable to most. That is why the resulting theory of relativity was so revolutionary that it was outright rejected out of hand by virtually all of those who considered it. Yet, it triumphed.

What it required was not only a new set of equations that governed observable phenomena, but a complete reworking of the fundamental concepts that we thought to be basic to the universe itself. One of Einstein's heroes, H. A. Lorentz, had proposed in his landmark paper "Electromagnetic Phenomena in a System Moving with Any Velocity Less than that of Light" [8] a first step toward the reconciliation of mechanics and electrodynamics that treated the distance between two objects as if it were not an objective fact of the world but a relative notion, relative to the state of motion of the observer. When an observer who is at rest measures the length of an object, she will come up with a number for its length. When a second observer who is in motion relative to the object measures the same object the moving observer will measure it as smaller, but only in the direction of motion. The object will have the same width and height, but depending on the speed of the observer, it will have a smaller length. The faster the observer, the smaller the length. As the observer approaches the speed of light, the length approaches zero. Lorentz's theory wherein objects behave as if they squish in the direction of motion is known as the Lorentz contraction.

Einstein took Lorentz's theory and did two things. First, he removed the "as if", so that there was no longer an objective fact in the world as to the length of an object. It is not as if the object contracts for moving observers, it does. Length thereby became a property that was frame-dependent. There is an objective fact about an object's length relative to a given frame of reference, but choose another frame and the length changes.

The second thing he did was to augment it with a similar equation for time. Duration was no longer an objective fact of the world. If an observer at rest measures the length of time between two

events, she will get a number. If another observer is in motion relative to the resting observer and measures the time between two events, he will measure a larger number. This is called "time dilation".

When we add the transformation equations that Lorentz derived for space with the new transformation equation for time from Einstein, the result is the special theory of relativity and it forces us to reconceptualize not only space and time, but motion, mass, and energy. We see the universe through completely different basic notions.

One effect of this is that we have a new relativized sense of time. In this sense, there is an absolute—causation. Causes always come before effects. If event A causes event B, then all observers will always agree that A came before B. Relativity respects causation in an absolute sense of time.

But for any two events A and B that are not causally related, that is, A does not cause B and B does not cause A, there is no longer an objective fact of the world about their time order. There will be one observer for whom A and B are simultaneous. There will be others for whom A happened before B, and yet others for whom B happened before A. Who is correct? All of them. The "relative" in relativity refers to the principle of relativity according to which there are no privileged observers, in Einstein's words, "the same laws of electrodynamics and optics will be valid for all frames of reference for which the equations of mechanics hold good [7] (pp. 37–38)". If a property depends upon the perspective from which it is viewed, then it is perspectivally true within that perspective; but, if there are other perspectives with different measurement outcomes, then there is no objective fact of the matter concerning that property.

All three of the properties of Newton's absolute time—absolute simultaneity, absolute time order, and absolute measures of duration—are in this way eliminated as objective facts of the world in this new understanding of the nature of time. Same time, time order, and amount of time now become properties of perspectives, not properties of objective reality. Time is different for different observers based on their relative states of motion. It is not just that it is "as if" it is different, it doesn't just appear to be different; time is different for different observers. Time is now a frame-dependent phenomenon and not an objective, unique feature of reality.

*1.4. McTaggart's A-Time vs. B-Time*

The final sense of time that needs to be understood for the following discussion derives from the work of James McTaggart Ellis McTaggart who distinguishes between what he terms A-time and B-time.

You are now ten years older than you were ten years ago. True, but trivial. But why is it true? Is it true because ten years of time have flowed past you or because you have moved through ten years of time? The former interpretation is the A view of time, whereas the second employs the B view of time.

In A-time, there are three metaphysically distinct regions of time.

> "Positions in time, as time appears to us *prima facie*, are distinguished in two ways. Each position is Earlier than some, and Later than some, of the other positions. And each position is either Past, Present, or Future. The distinctions of the former class are permanent, while those of the latter are not. If M is ever earlier than N, it is always earlier. But an event, which is now present, was future and will be past". [9] (p. 458)

In other words, there is a privileged instant, the now. Now is what is occurring. Once an instant has had its turn being the now, it becomes part of the past. The past is fixed. When an instant is awaiting its turn to be the now, it is in the future. Future instants are of a different sort than past instants because the content of the instant is open, unfixed, yet to be determined. There is, in A-time a metaphysical distinction to be drawn between past, present, and future and the status of each instant changes as it flows by. We are fixed and, in A-time, moments of time moves past us changing their nature as they go.

In B-time, on the other hand, time is a single unchanging sort of metaphysical entity. B-time, also known as block time, is set and fixed. It does not flow. It simply is as it is. We, on the other

hand, are moving through time, at least our consciousness is. We can remember the past but not the future because of the properties of ourselves as temporal beings, not because there is any metaphysical difference in the moments before or after any given instant of our consciousness. Everything that happens in time is fixed, fixed in time always and for all time. The future is not open, it is not undetermined, it is just not yet revealed to our consciousness. Time only appears to flow.

So, we have four distinct distinctions as related to time and film: film time vs. plot time, physical time vs. phenomenological time, classical Newtonian time vs. Einsteinian relativistic time, and McTaggart's A-time vs B-time. With respect to the first two distinctions, we are only interested in physical, film time. This allows us to set out two claims with respect to the other two distinctions: (I) Film time is restricted to classical B-time, and (II) Film time for SIIF video games does not have this restriction and thereby is capable of being relativistic A-time.

## 2. Film as Constrained to Classical A-Time

Not only is film a perfect model for the explanation of Newtonian absolute space and time, but with respect to absolute time, it is bound by it. Because by its nature, film is a linear stacking of absolute spaces into a well-defined, inviolate order, we see all three of objective facts of the universe that are the essential elements of classical Newtonian time: absolute simultaneity, absolute time order, and objective time intervals.

It is trivial to number the frames in order. There is a fact of the film as to what appears in each frame. So, to each event, that is, each thing in a frame, you can attach the number of the frame. If two events have the same number attached to them, then they are simultaneous. If not, then not. Hence, we have absolute simultaneity.

Similarly, for any two events, if the numbers are not the same, then whichever has the smaller number occurred first in film time. Thus, we have absolute time order. Subtract the smaller number from the larger and there is our absolute measure of duration. To say that film time is required to be classical Newtonian time is almost trivial.

Similarly, with respect to the requirement that film time must be B-time. It is true, in some sense that film time flows in that the film itself moves in the projector. But we see in the mechanics, exactly the B metaphysic described McTaggart. Each moment of time, that is, each frame, is fixed. The ending of the film while being watched is unknown to the consciousness of the observer, the viewer, but it is already fixed, printed on the celluloid.

One might object that since the frames can be divided into three categories—those that are on the uptake reel, the one that is in front of the projector's light and being projected onto the screen, and those that are on the original reel, that we can divide the film into the dynamic categories of A-time.

But the problem for such an attempt to render film time as A-time is that there is no metaphysical difference as the result of being projected. The essential metaphysical element in A-time is that when an instant has its turn as the present, that itself fixes the content of the moment as it moves into the past. The move from open future to experienced present to fixed past is the metaphysical mystery of time, a mystery only present in A-time not in the eternally fixed set of interrelated moments of B-time. The film has its moments fixed. The film itself is not affected by the act of the showing. The only change is in the consciousness of the viewer. Hence, we can conclude that physical film time must be classical, Newtonian B-time.

## 3. SIIF Video Games as Capable of Relativistic A-Time

This restriction, however, is removed when we move to the new approach in graphical narrative art, the SIIF video game. It is not our contention that this alternative temporality holds for all video games. It does not. Time in *Tetris* is straightforward Newtonian classical time. But, then, it is likely true, as Robson and Meskin point out, that such video games are likely not to be considered works of art [4] (p. 166).

We are not making general claims about all video games, but we do contend that at least some video games ought to be considered works of fictive art and we are only interested in those. It is the category of interactive video games that are the sort Robson and Meskin call SIIFs. These are works of graphical narrative art as much as films are.

One might object that these video games are not works of fiction. We agree with the grounds presented by Robson and Meskin that they are.

> "on the most popular contemporary accounts of fiction, which categorize something as fiction largely in virtue of authorial intentions to make an audience imagine or make believe certain contents, the vast majority of video games will plausibly count as fictions. Genre theories which appeal to a cluster of non-necessary features such as 'invented elements' will also, we suggest, do the same". [4] (p. 166)

As such, we should consider these games to be works of fiction, and so these video games are every bit as much art as a novel or film. Other elements—the moving images and music—are also surely artistic. So, when we consider a video game as a whole, it should be considered an integrated, multimedia work of art.

What is relevant about these works of art are two additional aspects. First, is the obvious one, interactivity. Where one is not merely a passive observer in a film, the viewer, for example, makes inferences or has expectations during the viewing, the content of the film is generally not about the viewer. There is a sense in which the viewer is allowed to glimpse the happenings of the fictional world, but remains alienated from it.

In a video game of the sort that we are interested in, however, the viewer is a participant who occupies a place in the fictional world through an avatar who not only observes the happenings of the fictional world, but participates in them, engaging with the fictional world in a fashion that changes it. This is different from, say, watching a book illustrated with short videos where you have to push an arrow to advance to the next page and again to start each page's video. This does not count as interactivity, since the fictional world does not change as a result of the actions of the reader.

The player of the game possesses a degree of autonomy in the playing. As such, the playing of the game is about them. As Robson and Meskin point out, this is supported by the linguistic patterns we use in describing the events in the playing of a game. A player of *Wipeout* would say "I made the other craft explode [4] (p. 168)" and a player of *Marvel vs. Capcom 3* would say "I beat Galactus by swinging him on my web". This self-attribution is not metaphorical. The player does claim the fictional achievement. As such, these are not merely works of fiction, but self-involving interactive fiction.

But while Robson and Meskin want to take SIIFs as a category and contend that they are the true object of insight in contemporary video game studies, we contend that there is something important in the video games of the category that does not hold for the other SIIFs. The key to the novel temporality is the explicitly ludic elements of the artform.

Structurally, these games possess, an original launch state and, either implicitly or explicitly, a failure state. What is crucial is that the player can lose. The failure state may be a simple "game over" screen or it can be more subtle. For example, a game may give the player a set of dialogue choices in a tense scenario where an ally is in danger. If the player chooses wrong, the ally dies, though the game may continue. In this case, the ally dying may constitute a failure state, even though play continues, if the player could only reach an outcome that is undesirable. Not all failure states have an element of finality, and many serve as simple temporary roadblocks where a player is forced to redo some portion of the game, but the ones of interest to us are the failure states with this degree of finality. These games are not only representations of fictional worlds, but because of the autonomy of the interactivity, they are essentially teleological experiences which require a development in which it is possible for the telos of the game to not be actualized within the playing of a token of the game.

There are interactive, non-teleological experiences such as virtual reality experiences where one can walk about a region and explore it. Such experiences are a form of interactive art in that the wearer

of the virtual reality apparatus can choose what to and what not to look at, but these choices have no impact on the course of the experience. This lack of ludic failure allows us to think of this aesthetic experience as an exploration of a space with no time, or potentially Newtonian time similar to films.

Similarly, the possession of a success state is also not sufficient for the sort of revised temporality we will argue exists in the SIIFs. Interactive video experiences can be crafted with a goal, but set up so that success is guaranteed. Such an experience may have a temporal element in that there are experiences that will be necessarily had along the way to the unavoidable success state, but this will not allow the alternate temporality because the fixed end state threatens to reduce the game time to B-time.

The arguments here ought to seem familiar as being similar to Karl Popper's arguments against the Logical Positivists [10]. Where the Logical Positivists sought unsuccessfully to use verification as a criterion of cognitive significance for scientific propositions, Popper showed that it is the possibility of falsification that, in fact, provides non-tautological, cognitive content for scientific propositions. In a similar way, we will show that the possibility of failure is essential for the video experience to have the new temporality.

What differentiates the sort of video games we want to explore from film and these other video game or game-like experiences is that because the player is autonomous and placed within the fictional world through an avatar, and because the game is implicitly teleological, but without a guarantee of actualizing the telos, there will a path from the starting state to the success or failure state such that the need to do what the player does in order to avoid failure creates the stream of instances that constitute the universe of that particular playing of the game.

We must distinguish the game from the playings of the game and the associated time. *Halo*, *Resident Evil 5*, and *Grand Theft Auto 4* are games. They are the works of art, the SIIFs. The engagement of the game by a player is a playing. The game creates the fictional world, populated with objects and events through the fictional space and time. The player's avatar is one of those things and is capable of causing events through interacting with the other fictional objects.

There will be events that are a part of the artwork itself independent of the playing, that is, there will be happenings that will occur in the fictional world that are not the result of the choice of the player. Following Walton, we will consider these elements of the "work time". We can lay out these fictional events in the fictional world on a timeline that is always a part of the fictional world, that is, they are a part of the work of art itself and so are to be considered elements in work time.

But events occur in the interstitial stretches between these events. These are the particular events of "game time", that is, they are true of the particularized fictional universe created by the interactive autonomy of the player engaging the work of art. When we order the events of game time within the timeline of events in work time, we have the full time of play.

This time is correlate of film time. But where we have seen that film time is restricted to Newtonian B-time, in the case of game time for video games that are SIIFs there is the possibility of the game time having the structure of relativistic A-time.

These SIIFs often have a sort of dual narrative structure, that is, where the game has an overarching central storyline, which we will refer to as the main quest, and a series of side stories, or side quests. While the main storyline often plays out in a predetermined fashion in work time, the side quests provide the ability for other events in the game world to take place in various different game times during the course of the story's unfolding.

An example of this structure is present in *Mass Effect 2* where the main story concerns your character's quest to stop the Reapers from wiping out all the sentient species of the Milky Way galaxy. Along the way, you have the opportunity to do several missions which make your allies more trusting of you and ultimately help determine if your allies survive or die. These missions can also be undertook in a variety of orders. In this case, these missions serve as events with narrative importance that occurs at different points in the timeline of different players, that is it occurs simultaneously with different events in different narrative in different playings. That is to say, events in the game world will not have absolute places in the timeline of the game world.

It is important to restate that the game is the type and the playings are tokens. The game is a work of art independent of the playings in the same way that the strip of celluloid is a work of art independent of the viewings. But unlike the viewer of the projection of the celluloid, each player's experience of the game is different. It is a single universe, but with multiple observers, each with a different frame of reference, viewing events in time, but experiencing them differently from their frame of reference.

Within these frames of reference, causation is absolutely preserved. Again, it will be the case that certain events in game time do not occupy fixed locations within the timeline of play time. But the alterations of time order will never violate causation. If a mission is necessary for completing a later task in order to reach a success state (such as making sure your crew survives), failing to complete this mission before the event will result in a failure state. Time order in the fictional universe of the work of art is no longer fixed, but the lack of absolute time order only holds for non-causally-related events.

So, let us take stock of the temporal relations in the fictional world of the SIIF video game and how they differ from temporal relations in the fictional world of the film. Most obviously, at the start of the film, the ending is already printed on the final frames. Film is necessarily B-time. But at the launching of the game, it is yet to be determined whether, or at least how, the player will lose, or if the player will win. The game's ending is not yet determined. Even in older non-SIIF games like *Space Invaders* or *Tetris*, where each successful phase it met by a more challenging phase, there is no success state, but there is an uncertainty about the nature of the failure state that will conclude play. In any given game, there may or may not be a success state, but either way there will not be a fact of the world about the nature of the final state that will end the game when the game first begins. As such, because of these ludic elements, video game time in general can be considered to be McTaggart's A-time. It is the universe of play that changes throughout and not merely the player's conscious experience of it.

In the specific case of SIIF video games, however, the A-time is the time of a narrative. In these games, there are events that affect each other. The causal structure underlying the narrative remains fixed. But from different reference frames, that is, from the experiences of different players playing different playings of the game, the non-causally related events will occur in different time orders. We have thus lost any sense of absolute simultaneity, time order, or duration in the author's created fictional universe. What we, therefore have is an artistic graphical representation employing relativistic A-time in the telling of the narrative of the SIIF game. Just as vanishing point perspective altered the ability of painters to portray a physical dimension, so video games do something markedly similar for graphical narrative art. It allows for a radical advance in the artistic representation of a physical dimension.

Video games are, in certain corners, seen as mere mass-produced entertainment for the great unwashed, not as an artistic medium to be taken seriously. Of course, the same was initially true of cinema. But as film has come to be taken seriously as a form of art, so too should video games.

Robson and Meskin do take some video games seriously as art, but only philosophically important as representatives of a larger class of artistic works, self-involving interactive fictions. These, they contend, is the actual site of insight of works in video game theory. But the artistic expansion of graphical narrative fiction to now include relativistic A-time as a possible temporality is not available to all other SIIFs.

If we consider tabletop RPGs like *Dungeons and Dragons*, it is certainly true that the dungeon master will have to do significant work to create the preconditions for the adventure. It is certainly true that in certain ways this work resembles that of the maker of the video game. But the resulting works are different. The dungeon master will have to be an interactive part of play. The dungeon master will have to make unforeseen choices in the course of the adventure. In the case of the video game, the world is fully self-contained. The video game is a fictional work of art. Can the same be said of the work of the dungeon master? Perhaps it can be seen along the lines of a jazz performance where a chart had been prepared as the composition and then improvised upon by the players. But this means that there is not the pre-existing absolute causal structure of the SIIF video game and thus the time

of play of the adventurers in the game of *Dungeons and Dragons* will be of a different sort than the relativistic A-time of the time of play of the playing of a similar SIIF video game.

In addition, while it may be possible to represent time in this way in other forms of SIIFs, to do so would require a severe change of the structure of those fictions, while the creation of a video game lends itself naturally to this type of structure and seems to be the natural vehicle for representing A-time in this way.

So, we disagree with Cova and Garcia and with Robson and Meskin. There is, on the basis of what we have argued here, reason to take SIIF video games as philosophical and aesthetically interesting works unto themselves. Indeed, we should see in them, a direct advancement of graphical narrative art.

**Author Contributions:** S.G.'s expertise in the foundations of physics led to the framing of the explications of the concepts of time, while J.R.'s expertise in technology and gaming led to the understanding of video games and role-playing games in general. The synthesis of these notions in the wider argument is the result of joint investigation.

**Funding:** This research received no external funding.

**Acknowledgments:** We would like to thank Vernon Cisney for providing the impetus for this discussion.

**Conflicts of Interest:** The authors declare no conflict of interest.

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
