# Peer review of "It’s about Time: Film, Video Games, and the Advancement of an Artform"

_philosophies, doi:10.3390/philosophies4040056_

Round 1

Reviewer 1 Report

The lack of citations in the article is surprising -- including citations would help with credibility, and to help readers understand what scholars have discussed before (for example, in terms of definitions for what constitutes a video game).

Examples in some of the more abstract descriptions could help readers understand the point (e.g., give us a couple well known movies as examples to help us understand the differences you discuss on p. 2).

The conclusion could use more fleshing out: I have a vague sense of this new importance of video games stemming from the main argument of the paper, but perhaps giving us more would help us really feel the impact. I'm not sure exactly what this could be -- maybe mention specific video games and how their use of time has furthered their status as artworks, so that we might say, oh yes I've played that or at least have heard something that game and yes of course its use of time is really important (e.g. in Braid, Life Is Strange, maybe...?). Right now, it's all in the abstract realm, and so it's difficult to feel the impact. I think you're onto something here, it just needs more color and grounding in things readers have some experience with to drive the point home for a wow moment at the end of the article.

Author Response

Point well taken.  In the rewrite we take care to tie the points to specific examples.  If this is about an artform, then it ought to hold for instances of the artform.  In the revision, we will embed the argument within the Waltonian/anti-Waltonian debate of Cova and Garcia and Robson and Meskin, arguing that this novel temporality (relativistic A-time) is specific to video games, thus making the category of video games (in virtue of their multiple endings as a result of player autonomy) something that makes the specific artform philosophically and easthetically interesting.  This should help focus the revised argument and locate it within the discourse better.

Reviewer 2 Report

The difficulty of identifying and defining video games as a medium against other, earlier forms has been a perennial concern of game studies, and coming at it through an overtly philosophical angle and through the dimension of time, this article makes a helpful and original contribution. McTaggart’s distinction between A-time and B-time is not one I had encountered before, but it does offer a largely convincing (though see below) way of distinguishing between the passive reception of temporal structures in film, and the active participation in temporality of video games. The overview of different interpretations of time was helpful and thoughtfully clear (though NB need for references), and indeed the use of film in illustrating Newtonian space time helps to cement the argument very firmly indeed.

You posit the argument in terms of the potentiality of video games (‘Film time for video games does not have this restriction and thereby is capable of being relativistic A-time’), and in this sense of games’ potential in the abstract the broad claim holds true.

However, I have concerns about your local arguments, which depend upon a very narrow conception of what video games are (or can be) in reality. Compared to film, games have a far greater range of affordances, structures, mechanics of interaction, and ways of constructing meaning (e.g. procedural versus traditional level design) or of meaning being perceived by a player (e.g. emergent versus scripted narrative), and yet the paper does not really reflect with this breadth or engage with the relevant research which does. This means that throughout the discussion it’s easy to find counterexamples that don’t fit the rules of temporal structure that you are trying to advance. To give some specifics:  

lines 297-302. This paragraph seems especially problematic. The claim that 'The game is a static entity, an unchanging block of code' is only broadly true: a game may receive downloaded content between or even sometimes during periods of play, players may mod or amend the game code via an in-play command line, and perhaps most importantly for procedurally generated games like No Man’s Sky the ‘unchanging block of code’ will give rise to outputs that are not wholly predictable or statically determined at the point play commences. The phrase 'But unlike the passive viewer of the projection of the celluloid, each player’s experience of the projected code is different' is surely making an excessively polarising distinction. It is not credible to claim that viewers of a film are 'passive' or that their experiences are not also different based on their individual background, context of viewing, whether they get up and have a toilet break etc. This is probably just an issue of wording, and what is really meant is that viewers of a film cannot influence the course of the film in any meaningful way - but this too has exceptions, such as the game-film crossover Bandersnatch. Following from this is the claim at 305-307 that ‘If saving the village is necessary for completing a later task in order to reach a success state, failing to save the village before the event will result in a failure state.’ This is not necessarily true. For example, consider a game like Skyrim (which is the sort of game you may have in mind in your simplified example of wizards and main and side quests). Modifying the example we might say that ‘If obtaining weapon of power 10 is necessary for killing a later enemy in order to reach a success state, failing to obtain weapon of power 10 before the event will result in a failure state’ – except that in the case of Skyrim (or many others) a player can, having failed to obtain weapon of power 10 at the earlier point, reduce the difficulty level at the point of encountering the later enemy such that their pre-existing weapon of power 5 is now permitted to defeat the same enemy. What this reveals is that temporal causation can be made to interact with other non-temporal and non-narrative aspects of the game such as a player’s attributes or possessions; with reference to the point above, the game’s code does not control temporal causality in a static way but might deliberately allow the causal rules to be modified mid-flow.   As soon as one considers genres like endless runners or even sandbox games like Minecraft in some play modes, general claims like 'it is the imperative of the player to avoid the failure state while interactively leaving an impact upon the universe of the game' or that ‘The video activity cannot be a game if the player cannot lose’ break down. And how do MMORPGs, which are often experienced not for purposes of engaging with narrative but for chat-based interaction, fit in? What about players of sandbox games who don’t play for the prescripted narrative but for other purposes? Nick Yee’s work on the multiple motivations of players could be a useful starting point.

In short, these examples suggest that your case for the temporal structure of games either needs to be made with reference to a more specific sub-genre of games where the temporal comparison does hold true (ironically, text-based adventures would be one that would probably work, as Aarseth’s Cybertext demonstrated two decades ago) or you need to acknowledge the diversity of the genre and, where necessary, dial back accordingly from overarching claims.

Other more particular areas to address should be:

The title, abstract and penultimate couple of paragraphs convey the impression that we 'need' the new medium of games because they allow for a 'more complex representation of time' or that they 'advance' the art form from our previous ‘best’ artistic representation of time. This opens all sorts of further questions as to how ‘best’ is defined. One could make a very convincing case (see e.g. Peter Brooks on Reading for the Plot) that it is precisely because other narrative forms are not able to represent time in a one-to-one relation that they have energy and interest. Great high modernist literature emerges from the imaginative struggle to represent new scientific understandings of time; the films of Quentin Tarantino are popular and aesthetically powerful precisely because they exploit and draw attention to the temporal strictures of the medium. The body of your paper doesn't actually make aesthetic value judgements, so to avoid misrepresentation I would advise rewording the abstract (and title) to position the paper as not encroaching on questions of value, only ontology. I would also revisit the final couple of paragraphs along the same lines. The paper is lacking any references and substantiation to existing literature (checking the Philosophies guidelines I assume these should be present). Readers would find it impossible to chase up, for example, the most relevant material defining the four approaches to time outlined in the first half. Discussion of story and film time would surely be linked to core narratological debates (e.g. Genette, Seymour Chatman). The definition of video games according to three criteria overlooks the body of debate in this field that would help to support this. To give one example, for the purpose of Philosophies journal it might make sense to look back to Karl Popper on falsification, but it would equally be appropriate to project forward to the ample work within contemporary game and play studies which identities the possibility of failure as a key criterion (obvious touchstones include Jesper Juul, Espen Aarseth, Ian Bogost). Or at 256 in making the distinction between games and VR Marie Laure-Ryan’s work on Narrative as Virtual Reality could be useful (though also interestingly complicating to your suggestion that VR is exploration). There appear to be several syntax/grammar issues that should be corrected (sampling more or less at random: 'Plot time may be explained to obeys law'; 'Perhaps the plot time has a circular, instead of linear[,] structure'; 'time order may is no longer fixed'; 'there will a path from').

As it stands at present, I feel like a reader coming to the paper from a background in philosophy (but no deep knowledge of games) will receive a hackneyed and monodimensional impression of what games are and how they operate; and they won’t be helped by the lack of references to the extant work that shows the actual diversity of the field. A reader coming to the paper from a background in game studies will immediately spot the flaws in the discussion by being able to conjure counterexamples. As a consequence, both will lose confidence in the interesting and original contribution of mapping different interpretations of time onto the genre, which would be of great value. I hope that this feedback is constructive and useful in helping to realise the benefits underpinning this paper.

Author Response

These are very helpful comments and will be accounted for in the rewrite.  We have reframed the argument in terms of the anti-Waltonian arguments of Cova and Garcia and Robson and Meskin, arguing now that there is, in fact, something philosophically and aesthetically important about certain video games (those that Robson and Meskin identify as Self-involving Interactive Fictions) qua video games.  It is the ludic elements that give rise to the novel temporality that was the heart of the original submission and thus we have grounds to distinguish video game SIIFs from the broader class of SIIFs that Robson and Meskin argue is the actual object of insight in the work of much philosophical work on video games.  We have reworked the sections that were found problematic and done so with references to specific works (films and games).  The comments and concerns were spot on and we hopefully have addressed the concerns in the rewrite.

Round 2

Reviewer 2 Report

This revised version is undoubtedly a more convincing article. The focus on SIIF, rather than video games in general, may be more modest but is much more watertight as a consequence. This is helped by the use of more specific examples of games, and indeed films. The introduction engages with some of the wider debates, in a way that more clearly positions the paper as speaking to games from a philosophical perspective, and with a reader from that discipline in mind, rather than from game studies as a whole (where many alternative attempts at definition have already played out). 

Two remaining concerns (hence indicating as requiring 'Minor Revision' rather than straight 'Accept') are:

Although engaging with more secondary literature, there still seem to be no in-text references as per house style (numbered square brackets to link to the reference list). https://www.mdpi.com/journal/philosophies/instructions#preparation. Perhaps the journal editors could advise as to whether this is expected?

The abstract could perhaps be tweaked to mention SIIF, given this (rather than 'video games' as a whole) is where the body now largely focuses.

Thanks for reworking this stimulating piece in relation to earlier comments in a thoughtful and deep-rooted way.

Author Response

We appreciate the input.  the changes have made this a much better paper.  As for the minor revisions...done.